# QSPR Modeling and Experimental Determination of the Antioxidant Activity of Some Polycyclic Compounds in the Radical-Chain Oxidation Reaction of Organic Substrates

**DOI:** 10.3390/molecules27196511

**Published:** 2022-10-02

**Authors:** Veronika Khairullina, Yuliya Martynova, Irina Safarova, Gulnaz Sharipova, Anatoly Gerchikov, Regina Limantseva, Rimma Savchenko

**Affiliations:** 1Faculty of Chemistry, Bashkir State University, 450076 Ufa, Russia; 2Institute of Petrochemistry and Catalysis of the Ufa Federal Research Center of the Russian Academy of Sciences, 450075 Ufa, Russia

**Keywords:** antioxidant activity, antioxidants, QSAR models, QSPR models, GUSAR2019 program, QNA descriptors, MNA descriptors

## Abstract

The present work addresses the quantitative structure–antioxidant activity relationship in a series of 148 sulfur-containing alkylphenols, natural phenols, chromane, betulonic and betulinic acids, and 20-hydroxyecdysone using GUSAR2019 software. Statistically significant valid models were constructed to predict the parameter logk_7_, where k_7_ is the rate constant for the oxidation chain termination by the antioxidant molecule. These results can be used to search for new potentially effective antioxidants in virtual libraries and databases and adequately predict logk_7_ for test samples. A combination of MNA- and QNA-descriptors with three whole molecule descriptors (topological length, topological volume, and lipophilicity) was used to develop six statistically significant valid consensus QSPR models, which have a satisfactory accuracy in predicting logk_7_ for training and test set structures: R^2^_TR_ > 0.6; Q^2^_TR_ > 0.5; R^2^_TS_ > 0.5. Our theoretical prediction of logk_7_ for antioxidants AO1 and AO2, based on consensus models agrees well with the experimental value of the measure in this paper. Thus, the descriptor calculation algorithms implemented in the GUSAR2019 software allowed us to model the kinetic parameters of the reactions underlying the liquid-phase oxidation of organic hydrocarbons.

## 1. Introduction

In the course of long evolution, people have surrounded themselves with a huge variety of chemical compounds, which are used in the process of life. A significant portion of these substances are organic compounds. During the use of these substances, their properties change under the influence of external conditions (temperature, solar radiation, and many others), thus reducing the quality of their performance. One of the most important processes leading to the deterioration of the performance characteristics is the oxidation by atmospheric (air) oxygen. This process, which follows a radical chain mechanism, can be carried out as auto-oxidation, where an oxygen molecule detaches a hydrogen atom from the weakest C-H bond and forms a primary radical, or has initiated the oxidation where initiators are present in the reaction medium that can easily initiate the oxidation process [1,2,3,4,5,6,7]. This creates a chain process in which labile, highly reactive, intermediates, such as radicals of a different nature and peroxides are formed [8,9,10]. As a result, the organic substrate undergoes a thermal-oxidative degradation and loses its functional characteristics.

To inhibit unwanted free-radical oxidation processes, minor additives of compounds, called antioxidants (AOs), are widely used. This is a group of various classes of organic compounds that can react with radical and peroxidation products and thus either inhibit or significantly slow down the development of free-radical processes in various systems capable of oxidation [1]. For the effective and targeted action of AOs, it is necessary to determine the quantitative characteristics in the form of rate constants for the steps of the mechanism responsible for the inhibition effect. The mechanism of the radical chain oxidation of organic compounds in the presence of AOs is very complex. Therefore, this requires considerable time and involvement of unique and expensive methods of physico-chemical experiments. Meanwhile, there are known methods of mathematical modeling that can be used to determine the necessary rate constants and do not require the investigation of the reaction mechanism. One such method is the quantitative structure–activity relationship (QSAR) / quantitative structure–property relationship (QSPR) modeling, which is widely used to identify the lead structures among the biologically active compounds, including drugs.

This approach is based on the assumption that the properties of chemical compounds are determined by their structure. The essence of the QSAR/QSPR methods is to describe the structures using correctly chosen descriptors and apply these descriptors in combination with mathematical and statistical methods to build valid QSAR/QSPR models focused on the reliable quantitative prediction of various types of biological activities and physico-chemical properties of organic compounds, respectively.

One of important advantages of the QSAR methods is that the physico-chemical properties and biological activities can be modeled on the basis of a relatively small number of training set structures (30 compounds).

There are quite a few monographs and papers describing the ideology of the QSAR/QSPR methods and software packages for their practical implementation, as well as the application of the QSAR/QSPR methodology to search for potential drugs and for the safety assessment of chemicals [11,12,13,14,15,16,17,18,19,20,21,22,23]. Numerous studies have shown that the inclusion of the QSAR approaches in the development of new hit and lead compounds can significantly reduce the time and material resources and provide for a targeted synthesis of the compounds possessing the required set of properties. Several types of classification of the QSAR/QSPR methods demanded by researchers have been described in the scientific literature. These types of classification are based on the choice of a set of descriptors and machine learning methods to construct mathematical equations [24,25,26,27,28,29,30,31,32,33,34,35,36,37]. Detailed considerations of these classifications as described, can be found, for example, in [24,29,38,39,40,41,42,43]. The most widespread of all known QSAR methods are those based on the structural formulas of chemical compounds (2D QSAR), as well as the methods that use a spatial description of the chemical structures (3D QSAR) [24,25,26,38,40]. The use of 3D-QSAR methods is justified for the quantitative analysis of the relationship between the structure and the enzymatic specificity of the biologically active compounds. When modeling the quantitative relationship between the structure and the physico-chemical characteristics, it is quite objective and exhaustive to use 2D QSAR methods.

In the last two decades, the research on antioxidants were concerned, apart from the widely known AOs, such as the sterically hindered phenols and aromatic amines, with the synthesis and experimental study of hybrid molecules, such as chromanol conjugates with lupanoic acids, tetrahydroquinoline, analogues of ecdysteroids with oxygen-containing heteroatoms in the steroidal backbone. Owing to the presence of the pharmacophore moieties, these compounds are promising as potential biologically active additives with a wide spectrum of biological activity. In addition, various research groups are currently synthesizing the structural analogues of these compounds. The rational design and synthesis of these compounds using modern virtual screening techniques, including QSAR/QSPR modeling, pharmacophore search, and molecular docking, occupies a crucial place among the strategies for selecting the directions for the chemical modification of the biologically active substances. The choice of one of these methods depends mainly on the goals of the study. If the ultimate goal of the synthesis is to obtain biologically active substances for further *in vitro* studies without involving enzymatic systems, then one of the 2D-QSAR/QSPR methods would be the most preferable choice in this case.

The name “2D-QSAR/QSPR” stands for the development of QSAR/QSPR models using 2D descriptors. Two-dimensional descriptors are widely employed in QSAR/QSPR modeling owing to the relatively simple calculation algorithms, based on mathematical equations. This method possesses a reproducible operability and does not require large amounts of time and computational resources. Furthermore, 2D descriptors make a huge contribution to the extraction of chemical attributes, and they can also represent, to some extent, three-dimensional molecular features. However, they should by no means be considered as final, since they often suffer from mutual correlation problems, insufficient chemical information, and the lack of interpretation [44].

The benefits of these methods for modeling the reactivity of oxidation inhibitors (antioxidants) in liquid-phase reactions are obvious both for the above-mentioned rational design and from the ideological standpoint, considering kinetic experiment techniques. In particular, reactant molecules and intermediate radicals formed from the reactants are uniformly distributed throughout the reaction system; the reaction system is a homogeneous solution; the liquid-phase oxidation reaction of organic substrates takes place virtually in the whole reaction system; the reaction rate in a homogeneous solution (reaction mixture) is actually determined by the frequency of collisions of the reactants with one another, the solvent nature, and the oxidation substrate; in the reaction medium, the substrate is oxidized by the atmospheric oxygen; the reaction is started by the initiators in the absence of enzymes.

In this connection, before carrying out a synthesis and experimental studies of the antioxidant activity (AOA) of chromanol conjugates with steroidal and nonsteroidal compounds in non-enzymatic model systems, it is advisable to quantify their AOA using the QSPR methodology. Inclusion of these and other hybrid molecules in the training sets in QSPR modeling will expand the range of applicability of QSPR models focused on the prediction of the logk_7_ parameter in the series of phenolic antioxidants [45,46,47,48,49,50,51,52,53].

One of the programs used to calculate the physico-chemical and structural descriptors, to select the most significant of them, and to build consensus QSPR models based on them is the GUSAR2019 (General Unrestricted Structure Activity Relationships) program and its earlier versions GUSAR2013 and GUSAR2011 [11,12,17,54,55,56,57,58,59,60]. This program has proven efficient in the modeling of various types of biological activities for some heterogeneous organic compounds [54,55,56,57,61,62]. In our pioneering work [62,63,64,65,66], we reported the use of an earlier version of the program, GUSAR2013, for the QSPR modeling of antioxidants in the series of some phenols, amines, uracils, benzopyrans, and benzofurans. In doing so, the statistically significant valid models were constructed to predict the oxidation chain termination rate constants of logk_7_, in order to search for new potentially effective antioxidants in virtual libraries and databases. However, the training sets used in our previous models for predicting logk_7_ for phenolic antioxidants did not contain the structures of the chromanol conjugates with lupanoic acids, 20-hydroxyecdysone, and for this reason, they could not be used for the quantitative AOA prediction for the structural analogues of these organic compounds.

The main goal of the present study is to develop statistically significant valid QSPR models for the prediction of the parameter logk_7_ for biologically active phenolic antioxidants with general formulas **I**–**VIII** (Figure 1) (k_7_ is the rate constant for chain termination by the antioxidant molecule and is actually an objective quantitative characteristic of AOA) and to predict and experimentally determine this quantitative AOA parameter for two promising antioxidants, chromane derivatives. The practical significance of this study should be the applicability of these QSPR models to predict logk_7_ for the biologically active phenolic derivatives and, hence, an objective selection of such compounds as oxidation inhibitors from virtual and synthetic libraries and databases.

## 2. Results and Discussion

### 2.1. Prediction of the Numerical Values of the Parameter k_7_ Using the GUSAR2019 Program

According to the consensus approach implemented in the GUSAR2019 program, six consensus QSPR models M1–M6 were built to predict the numerical values of logk_7_ for the phenolic type antioxidants, namely the sulfur-containing alkylphenols, the natural phenols, the chromane derivatives, the betulonic and betulinic acids, and 20-hydroxyecdysone. These models differ in the type of descriptors they contain and the number of partial regression relationships. The descriptive power characteristics of the M1–M6 consensus models, calculated automatically in the GUSAR2019 program by comparing the experimental logk_7_ values with those predicted by these six models are presented in Table 1. Note that the determination coefficients, the standard deviations, and Fisher’s criterion values presented in Table 1 are the average values obtained taking into account all partial regression models included in the consensus model Mi (i = 1–6).

For a QSPR model to be adequate, that is, to be acceptable for use within its reach, its results must correctly describe and predict the target property. According to the recommendations of the QSAR/QSPR modeling experts, the validation is the most important concept in the development and application of the QSPR models. The validation of the developed QSPR models based on the external test set structures is the “gold standard” that validates the reliability of these models along with the acceptability of each step during their development: the assessment of the input data quality, the diversity of the data sets, the predictivity, the domain of applicability, and interpretability.

In this regard, to objectively characterize the descriptive and predictive powers of the M1–M6 consensus models, we performed the prediction of the logk_7_ values for the antioxidant structures contained in test sets TR1 and TR2. As in our previous studies [67,68,69,70], in addition to the parameters calculated in the GUSAR2019 program (average R^2^, average Q^2^, average F), we used metrics based on the R^2^ determination factors (R^2^, R^2^_0_, R^2’^_0_, average R^2^_m_, ΔR^2^_m_ Q^2^_F1_, Q^2^_F2_, CCC); and the metrics designed to estimate the prediction errors of the logk_7_ values (RMSE, MAE, SD) [34,35,36,37]. The metrics based on the prediction error estimates were used to determine the true prediction quality index for the parameter logk_7_ for the compounds of both test sets. Their calculation was performed using the Xternal Validation Plus 1.2 program [71]. The same program was used to check the models for systematic errors. 

The statistical criteria measuring the descriptive and predictive powers of the M1–M6 QSPR models, which were estimated for 95% of the structures of the training and test sets TR1, TR2 and TS1, TS2, respectively, are presented in Table 2, Table 3. Appendix A (Appendix A) present a complete set of the statistical parameters calculated using the Xternal Validation Plus 1.2 software for the training and test sets TR1, TR2 and TS1, TS2, taking into account both 100% and 95% of the antioxidant structures they contain.

The analysis of the statistical characteristics of the M1-M6 consensus models, summarized in Table 2, Table 3 showed that almost all of the models successfully reproduced the experimental data contained in both training sets (the condition is satisfied for 100% and 95% of the data). Thus, the values of the determination coefficients R^2^, R^2^_0_, R^2’^_0_, average R^2^_m_, and CCC, evaluated by comparing the values of logk_7_^pred^ and logk_7_^exp^ fully met all of the the requirements, corresponding to the models with a high descriptive power listed in part 2.3. The M6 model (100% and 95% of the data) had the highest descriptive power in a number of determination coefficients (R^2^, R^2^_0_, CCC). At the same time, other criteria indicated the best reproducibility of the experimental data of test sets TS1 and TS2 (100% and 95% data) using the models M3 (R^2’^_0_, ∆R^2^_m_), M4 (average R^2^_m_), and M5 (R^2’^_0_, average R^2^_m_). The M3 and M6 models were characterized by the lowest values of the prediction errors of the logk_7_ value for the structures of both training sets (RMSE, MAE, SD) at 100% and 95% of the data contained in them. At the same time, the best characteristics were in the M3 model (Table 2). The minimum SD value at 100% of the data in both training sets was shown by the M4 model. In the case of 95% of the data in these training sets, the best result was observed for the M6 model. Since the numerical values of the MAE for all of the models were in the range of 0.0599–0.0679, which is significantly lower than 0.706 (10% of the range of the simulated logk_7_ values), and simultaneously, the numerical values of the MAE+3SD criterion were also significantly smaller than 0.706, we can conclude that almost all of the models had a high descriptive power.

However, the same models were characterized by the rather low values of the different determination coefficients for the comparison of the experimental and predicted logk_7_ values for the 100% antioxidant structures contained in test sets TS1 and TS2. Thus, the coefficient of determination R_2_ and its analogs (R^2^_0_, R^2’^_0_) were in the range of 0.4500–0.6882, the CCC criterion ranged from 0.6483 to 0.8086, which allowed us to characterize the prognostic ability of these models as low. At the same time, the most successful predictions, if we focus on these criteria, were observed for the structures of test set TS2. Meanwhile, a more reliable estimate of the predictive power of the M1–M6 models, taking into account 100% of the data in the test sets, can be obtained by analyzing the criteria based on the logk_7_ prediction errors for the same antioxidant structures. Specifically, the MAE and MAE+3SD criteria ranged from 0.3472 (M6, TS2) to 0.4696 (M5, TS1) and from 1.4586 (M4, TS2) to 2.1112 (M5, TS1). According to these criteria, the models with moderate predictive powers are M1 (TS1), M3 (TS1), and M4–M6 (TS2). Thus, the analysis of prediction errors for antioxidant structures contained in test sets TS1 (100% data) and TS2 (100% data) did not remove the uncertainty factor in assessing the predictive power of the M1–M6 models.

The removal of 5% of the structures from both test sets led to a significant increase in the numerical values of the various types of determination coefficients and a decrease in the logk_7_ prediction errors for the structures contained in TS1 and TS2.

The numerical value of the R^2^ criterion increased approximately by 30% and ranged from 0.7289 to 0.8204. The coefficient of determination R^2^_0_ increased in parallel and was almost in the same range: 0.7263–0.8115. The maximum values of these criteria were found in both cases when the M1 model was used for the prediction tasks in the series of antioxidants contained in test set TS1 (95% of the data). According to the criteria mentioned in Part 2.3, the M5 and M6 models were insignificantly inferior in their predictive power. This fact was established in the prediction of logk_7_ for the antioxidants included in test set TS2 (95% data). From the analysis of the numerical values of all of the other types of determination coefficients, which are presented in Table 3, we can conclude that in some cases, the M5 model demonstrated the greatest prognostic ability. We reached this conclusion by analyzing CCC, R^2’^_0_, and average R^2^_m_ values for the compounds of test set TS2 (95% data). The highest values of the criteria Q^2^_F1_, Q^2^_F2_ differed in the results of the prediction of logk_7_ for the structures of the same test set performed using the M6 model (Table 3). When evaluating the prognostic ability of the M1-M6 models, taking into account the prediction errors of the logk_7_ values for 95% of the data in test sets TS1, TS2, the most successful predictions were also observed for the test set TS2 structures. The M6 model showed the lowest values of the RMSEP error, the SD standard deviation, and the MAE+3SD criterion. On the same dataset, the M5 model showed the minimum MAE error.

Thus, relying on the set of criteria summarized in Table 3, we can conclude that all of the models had a moderate predictive power in predicting the logk_7_ values for the antioxidant structures contained in test sets TS1 and TS2. An obvious proof of this fact is the plot depicted in Figure 2, which shows a satisfactory correlation between the experimental and predicted values of logk_7_ for the structures of test sets TS1 and TS2 (95% data).

The insignificant difference between the numerical values of the various types of determination coefficients, in combination with the acceptable values of the MAE and MAE+3SD parameters, summarized in Table 2, Table 3, indicates that the valid QSPR models focused on predicting logk_7_ values for the antioxidants can be constructed using either one particular type of descriptor (QNA or MNA descriptors) or a combination of the descriptors in a consensus approach.

Subsequently, the M1–M6 consensus model was used to predict the numerical values of logk_7_ for the antioxidants AO1 and AO2. The results of these calculations are summarized in Table 4.

The approximate 95% confidence interval for predicting future data is ±2RMSE if the model is correct and the errors are normally distributed.

### 2.2. Experimental Determination of the Inhibition Rate Constants k_7_ for Compounds AO1 and AO2. Methods of the Kinetic Experiment to Determine the Antioxidant Activity of Compounds AO1 and AO2

The synthesis, the physico-chemical properties, and the antioxidant assays of compounds AO1 and AO2 (Figure 3) were reported previously [72]. In the present study, we describe the kinetics of the radical chain oxidation of an organic compound in the presence of additives AO1 and AO2.

The experimental logk_7_ values for compounds AO1 and AO2 were determined by the manometric method using air oxygen absorption as a model liquid-phase oxidation of 1,4-dioxane, initiated by azobis(isobutyronitrile) (AIBN). The experiments were performed according to the standard technique described earlier [72,73,74,75,76,77,78]. The model reaction was carried out in a thermostatically controlled glass reactor where the solutions of the initiator (AIBN) and the studied substances in 1,4-dioxane were loaded. The temperature of the reaction mixture was 348 K. The reaction mixture was maintained in the thermostat for 5 min. The kinetic curves was measured using a universal manometric differential unit, the design of which was reported earlier [75,76,77,78]. Subsequently, the initial rates of the oxidation of 1,4-dioxane were calculated from the initial sections of the kinetic curves recorded in the absence and in the presence of compounds AO1 and AO2 using the least-squares method. The numerical values of the effective inhibition rate constants for compounds AO1 and AO2 were calculated from the degree of the decrease in the initial oxygen uptake rate during the oxidation of 1,4-dioxane. The initiation rate of the oxidative process was constant and was V_i_ = 1 × 10^−7^ mol·l^−1^·s^−1^. It was determined using the equation V_i_ = 2ek_p_[AIBN], where k_p_ is the rate constant of the AIBN decay, e is the probability of the radical escape into the bulk). For k_p_, the value measured in cyclohexanol was taken [79]:logk_p_ = 17.70 − 35/(4.575T·10^−3^), e = 0.5(1)

Since the reaction was performed according to the standard technique [73,74,75,76,77,78], we assumed that the initiated oxidation of 1,4-dioxane proceeded by the radical chain mechanism, which we schematically show in Figure 3 [1,2,3,4,5,6,7].

The antioxidant properties of AO1 and AO2 were studied in the AIBN-initiated radical chain oxidation of 1,4-dioxane in the kinetic regime at 348 K. The typical kinetic curves of the oxygen uptake in the presence of additives of AO1 and AO2 at different concentrations are shown in Figure 4 and Figure 5. In the absence of compounds AO1 and AO2, the kinetic curves of the oxygen uptake in the oxidation of 1,4-dioxane were straight lines, i.e., the reaction order with respect to oxygen was zero. Consequently, the oxidation of 1,4-dioxane proceeded in the kinetic regime. In this case, the chain propagation and termination reactions were run by peroxyl radicals.

As can be seen in Figure 4 and Figure 5, the introduction of the additives of AO1 and AO2 brings about a clear induction period in the kinetic curves of the oxygen uptake, indicating a pronounced antioxidant effect of the studied substances.

Using the Excel 2016 word-processor, we calculated the initial oxidation rates of the model substrate at different concentrations of the added substances. The resulting numerical values are presented in Table 5. As can be seen, the introduction of compounds AO1 and AO2 separately in the concentration range of (0.44–3.13) × 10^−6^ mol/L for AO1 or AO2, respectively, into 1,4-dioxane being oxidized, led to a decrease in the initial oxidation rate. Thus, the qualitative analysis allows us to conclude that we consider that both compounds effectively inhibit the oxidation process of the model substrate (Figure 4 and Figure 5, Table 5).

The numerical values of the effective rate constants of inhibition fk_7_ for each of the antioxidants were calculated using Equation (2). The condition for the applicability of this equation is a linear dependence of the inhibition parameter F on the concentration of the antioxidants. As can be seen from Figure 6, in the oxidation chain regime in the (0.44–3.13) × 10^−6^ mol/L concentration range of the AO1 and AO2 compounds, the inhibition parameter F, calculated from the initial rates of the inhibited oxidation of 1,4-dioxane by formula (2) actually followed a linear dependence on the AO1 and AO2 concentrations (Figure 6):(2)$$F=\frac{V_{0}}{V}-\frac{V}{V_{0}}=\frac{fk_{7}[InH]}{\sqrt{2k_{6}V_{i}}},$$
where V_0_ and V are the initial rates of the oxygen uptake during the oxidation of 1,4-dioxane in the absence and in the presence of each of the antioxidants taken separately, respectively, [AO] is the concentration of the added AO, k_7_ and 2k_6_ are the rate constants of the oxidation chain termination by the antioxidant and the quadratic chain termination via peroxyl radicals of the substrate, respectively [1,2,3,4,5,6,7], [RH] is the concentration of 1,4-dioxane ([RH] = 11.75 mol/L), k_2_ is the rate constant of the chain propagation for the oxidation of the model substrate (k_2_ = 7.9 l·mol^−1^·s^−1^ [2]). When calculating these values, we used the quadratic chain termination rate constant 2k_6_ = 6.67 × 10^7^ l·mol^−1^·s^−1^ known from the literature [2]. The errors in determining the fk_7_ and f values were calculated using the Excel 2016 word processor (Regression tab).

The adjustment of the experimental data in the coordinates of Equation (2), the effective inhibition constants for compounds AO1 and AO2 were determined to be f = (1.32 ± 0.3) × 10^6^ M^−1^s^−1^ and f = (1.08 ± 0.2) × 10^6^ M^−1^s^−1^, respectively. In addition, to determine the numerical value of the stoichiometric inhibition coefficient, we studied the dependence of the induction period, which appeared on the kinetic curves of the oxygen uptake, on the concentrations of AO1 and AO2. As can be seen from Figure 7, the dependence of the induction period τ on the concentrations of AO1 and AO2 is linear. In this case, it is correct to use Equation (3) to determine the stoichiometric inhibition coefficient f:(3)$$\tau =\frac{f[InH]}{V_{i}},$$
where τ is the induction period on the kinetic curves of the oxygen uptake during the oxidation of 1,4-dioxane inhibited by AO1 and AO2; V_i_ is the initiation rate of the oxidation. Conversion of the experimental data in the coordinates of Equation (3) gave the stoichiometric inhibition factors f for the antioxidants AO1 and AO2 to be 30 ± 4 and 40 ± 2, respectively.

The inhibition rate constant k_7_^exp^ for AO1 and AO2 was calculated by formula (4):(4)$$k_{7}^{\exp}=fk_{7}^{\exp}/f,$$

The numerical values of k_7_^exp^ for the antioxidants AO1 and AO2 were k_7_^exp^ = (4.3 ± 1.0) × 10^4^ M^−1^s^−1^ and k_7_^exp^ = (2.7 ± 0.5) × 10^4^ M^−1^s^−1^, respectively.

The comparative analysis of the calculated logk_7_^pred^ and the experimental logk_7_^exp^ values for compounds AO1 and AO2 (Table 4) suggests that the M1–M6 QSPR consensus model has a moderate predictive ability and can be applied to the search and development of new antioxidants. The difference between the predicted and experimentally determined logk_7_ values for these antioxidants does not exceed the 2RMSEP range.

Thus, all M1–M6 QSPR consensus models are characterized by a high descriptive and moderate predictive power for comparing the experimental and predicted logk_7_ values for training set structures TR1 and TR2, the external and internal test set structures TS1 and TS2, and compounds AO1 and AO2. These models can be used for the screening of virtual libraries and databases in order to search for new antioxidants in the series of some sulfur-containing alkylphenols, natural phenols, chromane and lupanoic acids, betulonic and betulinic acids, and 20-hydroxyecdysone.

In general, the approach implemented in the GUSAR2019 program, which was previously used only for modeling the biological activity of low-molecular-weight compounds, allows a high degree of reliability in modeling the kinetic characteristics of antioxidants expressed as the k_7_ parameter. Thus, this program can be recommended as an additional tool in the search for new antioxidants.

## 3. Research Methods

The simulation procedure was performed for the compounds whose formulas are shown in Figure 1.

### 3.1. The Methodology of the Computational Experiment

The QSPR modeling of the derivatives of the sulfur-containing alkylphenols, natural phenols, chromane and lupane acids, betulonic and betulinic acids, and 20-hydroxyecdysone with general structural formulas **I**–**VIII** (Figure 1) was performed using the GUSAR2019 (General Unrestricted Structure Activity Relationships) software [54,55,56,57,58,59,60,61,62].

The QSPR models were built in several stages, schematically presented in Figure 8.

### 3.2. Formation of the Training and Test Sets

The training sets TR1, TR2 and test sets TS1, TS2 were based on the array of S1 structures in accordance with the procedure described in our earlier studies [67,68,69,70,76,77,78,79,80,81,82,83,84]. This procedure reflects the rational separation strategy and is presented in Figure 9. The array of the set S1 structures included 148 sulfur-containing alkylphenols, natural phenols, hybrid molecules (conjugates of chromane and lupanoic acids, betulonic and betulinic acids, 20-hydroxyecdysone) with their corresponding logk_7_ values.

The parameter logk_7_ was obtained by taking logarithms of the numerical values of the inhibition rate constant k_7_ for the simulated antioxidants, which were measured experimentally and reported in the literature [67,68,69,70,76,77,78,79,80,81,82,83,84]. In fact, the inhibition rate constant k_7_, which we chose as the simulated parameter, reflects the specific rate of the inhibition of the liquid-phase oxidation of the organic substrates similar in oxidative capacity by the antioxidants. In modeling, it was assumed that the oxidation of the organic substrates in the presence of antioxidants, proceeds in several steps and can be schematically described by the following key steps, which have been studied in detail and described in the literature [1,2,3,4,5,6,7] (Figure 10).

Reactions 1 and 2 are the elementary steps of the oxidation chain propagation, reactions 6 and 7 are chain termination steps via the recombination of the peroxyl radicals RO_2_^•^ and via the antioxidant molecule, respectively. The antioxidant effect of the antioxidants included in the S1 data array is implemented through their reaction with the peroxyl radical of the RO^2-^ oxidation substrate RO_2_^•^. As a result, the peroxyl radical active in the chain propagation reaction is replaced by an inactive antioxidant radical. This is the AOA mechanism of the simulated compounds. Obviously, the higher the numerical value of the inhibition rate constant k_7_, the more pronounced the antioxidant properties of the organic compound.

The M1–M3 QSPR models were constructed based on the training set TR1, which included 123 antioxidant structures with their corresponding logk_7_ values. To test the predictive power of the M1–M3 models, we used test set TS1, which contained 25 antioxidant structures with their corresponding logk_7_ values. Both of these sets were derived by a 5:1 split of the S1 data set by transferring every sixth compound from S1 to TS1. The remaining 123 antioxidant structures were used to form the training set TR1. Preliminarily, all structures of the data array S1 were ranked in ascending order of the numerical value of logk_7_.

The training set TR2 included 103 antioxidants with their respective logk_7_ values and was designed to build the M4–M6 QSPR models. The validity of the M4–M6 QSPR models was tested using test set TS2. Both TR2 and TS2 sets were formed on the basis of the training set TR1. In this case, the TR1 set was subjected to a 5:1 split, with the transfer of every sixth compound from TR1 to TS2. The characteristics of the training sets TR1 and TR2 and test sets TS1 and TS2 are presented in Table 6 and Table 7, respectively. The data of these tables indicate that the compounds of the training and test sets are fairly evenly distributed over the entire range of the logk_7_ variability. At the same time, the AOA of the compounds of the TR1 and TR2 sets varies over a wide range (∆logk_7_ = 7.06). The range of variability of logk_7_ for the compounds of the test sets does not go beyond the range Δlogk_7_ = 7.06. In addition, as can be seen from Figure 1, the training sets are characterized by a high degree of molecular diversity. These conditions are important for building high-quality QSPR models and the correct forecasts based on them [34].

The compound structures of the training and test sets TR1, TR2 and TS1, TS2 were built using the MarvinSketch 17.22.0 software [85], and then were converted into the SDF format using DiscoveryStudioVisualiser [86].

### 3.3. Building QSPR Models

The M1–M6 QSPR models were built based on two types of substructural descriptors of atomic neighborhoods: QNA (quantitative neighbourhoods of atoms) and MNA (multilevel neighbourhoods of atoms) [11,12,17,54,55,56,57,58,59,60,61,62]. The calculation of these types of descriptors in the GUSAR2019 program was performed automatically from the structural formulas of chemical compounds, taking into account the valence and partial charges of all atoms. The specific features of the communication types were not taken into account in the calculations. The ideology of calculating the QNA and MNA descriptors is described in detail in the Appendix A and in previous publications [11,12,17,54,55,56,57,58,59,60,61,62]. However, the QNA descriptors cannot be physically interpreted due to the peculiarities of their calculation. In this regard, they are not explicitly displayed under the calculations.

The MNA descriptors are computed using the PASS algorithm (prediction of activity spectra for substances) [17,60], which predicts approximately 6400 “biological activities” with an accuracy threshold of an average prediction of at least 95%. These descriptors are generated based on the structural formulas of the chemical compounds without using any pre-compiled list of the structural fragments [11,17,60,87]. They are generated as a recursively defined sequence:Zero-level MNA descriptor for each atom is the mark A of the atom itself;Any next-level MNA descriptor for the atom is the substructure notation A (D_1_D_2_…D_i_…), where D_i_ is the previous-level MNA descriptor for i–th immediate neighbor of the atom A.

The neighbor descriptors D_1_D_2_…D_i_… are arranged in a unique manner. This may be, for example, a lexicographic sequence. The MNA descriptors are generated using an iterative procedure, which results in the formation of structural descriptors that include the first, second, etc. neighborhoods of each atom. The label contains not only information about the type of atom, but also additional information about its belonging to a cyclic or acyclic system, etc.

The QSPR model additionally included three descriptors of the whole molecule (topological length, topological volume, and lipophilicity), which were also calculated automatically in the selected program.

To reduce the descriptor space and select the most significant descriptors, we used the approach referred to as Both, in the GUSAR2019 program. This approach is new. It is proposed by the developers of the GUSAR2019 program and combines the simultaneous use of the two methods of the descriptor space reduction previously proposed by the same authors: the method of self-consistent regression (SCR), and its combination with radial basis functions (RBF-SCR). A detailed description of this method can be found in the Appendix A and in the relevant publication [60].

The developers of the GUSAR2019 program recommend using the SCR-RBF method to select the descriptors when the training set contains structurally heterogeneous compounds.

The stability of the constructed models was tested using a sliding control procedure with a 20-fold randomized outlier of 20% of the compounds from the training sets TR1 and TR2. Both procedures in the GUSAR2019 are implemented automatically [11,12,17,54,55,56,57,58,59,60,61,62].

The four final QSPR models, M1, M2, M4, and M5, were constructed using a consensus approach and included 20 partial regression relationships. The condition for combining several regression equations into one consensus model was their general similarity. The M1 and M4 models were constructed based on the QNA descriptors and three descriptors reflecting the topological length, topological volume, and lipophilicity of the simulated antioxidant structures. The M2 and M5 models were constructed according to a similar principle, but based on the MNA descriptors with the automatic addition of the same three whole molecule descriptors described above. The M3 and M6 models were constructed according to a similar principle, but each of these models included 320 partial regression relationships. At the same time, each of these 320 single models included in the M3 and M6 consensus models was constructed independently of each other, based on the three whole-molecule descriptors described above with the addition of either the QNA or MNA descriptors. Due to specific features of the calculation, the QNA and MNA descriptors do not lend themselves to an unambiguous physical interpretation. In this regard, the regression equations based on them are not displayed explicitly in the GUSAR2019 program. The final prediction of the numerical value of logk_7_ for a particular compound using a particular model was formed based on the results of averaging the predicted logk_7_ values of the single regression QSPR models included in this consensus model.

### 3.4. Assessment of the Descriptive and Predictive Powers of the QSPR Models

In order to ensure the consistency of the results, the same standard parameters were chosen to assess the descriptive and predictive powers of the M1–M6 consensus models. The descriptive power of the M1–M6 models was evaluated using metrics based on the determination coefficients R^2^ (R^2^, R^2^_0_, R^2’^, average R^2^_m_, CCC) and the metrics evaluating the prediction errors of the logk_7_ values (root mean square error (RMSE), mean absolute error (MAE), standard deviation (SD)) [34,35,36,37]. These statistical parameters were calculated using Xternal Validation Plus 1.2 for 100% and 95% of the data (to account for the outliers) in the training and test sets [88]. The Appendix A provides the formulas by which these criteria are calculated in this program. The internal validation of the M1–M6 models was performed using LMO cross-validation (Q^2^_LMO_) with a 20-fold exclusion of 20% of the compounds from the training sets.

Additionally, the predictive power of the consensus QSPR models was evaluated by comparing their predicted logk_7_ values with the experimental values of the same parameter for the new promising antioxidants AO1 and AO2, which were not included in the S1 data set (Figure 10).

The threshold values of the validation criteria for the above parameters for models of the high descriptive and predictive powers were as follows:For 95% of the data of the training set TRi, the numerical values of the determination coefficients R^2^, R^2^_0_, R^2’^_0_, and the CCC criterion should be close to each other and tend to unite;Numerical value of the criterion R^2^_m_ > 0.85 with ΔR^2^_m_ < 0.15;Numerical value of the average absolute error MAE should not exceed 10% of the activity range Δlogk_7_ of the simulated training set TRi;MAE+3SD parameter value (where SD is standard deviation) should not exceed 10% of the activity range Δlogk_7_ of the simulated training set TRi;Numerical values of the determination coefficients Q^2^_F1_, Q^2^_F2_ (calculated for the test sets) should be close to each other and tend to unite.

The quality of the QSPR models was considered low if they met the following criteria: For 95% of the data of the training sample Tri, the numerical values of the determination coefficients R^2^, R^2^_0_, R^2’^_0_, and the CCC criterion should not exceed the threshold value 0.6;Numerical value of R^2^_m_ ≤ 0.5 with ΔR^2^_m_ ≤ 0.2;Numerical value of the mean absolute error of the MAE exceeded 20% of the activity interval of the Δlgk_7_ compounds simulated by the training sample TRi;The value of the MAE+3SD parameter exceeded 25% of the activity interval of the lgk_7_ compounds simulated by the training sample TRi;Numerical values of the determination coefficients Q^2^_F1_ < 0.70, Q^2^_F2_ < 0.70 (calculated for the test sets) should be less than 0.70.

In all other cases, the descriptive and predictive powers of the models were evaluated as moderate, according to the criteria described above.

## 4. Conclusions

The QSPR strategy implemented in the GUSAR 2019 program was used to establish a quantitative structure–antioxidant activity relationship for a series of 148 sulfur-containing alkylphenols, natural phenols, chromane, betulonic and betulinic acids, and 20-hydroxyecdysone with the general structural formulas **I**–**VIII**. Six statistically significant valid QSPR consensus models were built. The models demonstrated a satisfactory predictive accuracy in predicting the parameter logk7 for training and test set structures: R^2^TR > 0.6; Q2TR > 0.5; R2TS > 0.5. All models showed a high performance, as they reproduced the known experimental data for the training sets with a high degree of accuracy. The cross-validation with a 20-fold exclusion of 20% of the training set data also showed good results. The validation of the prediction of logk7 by the estimation of these parameters for the compounds of two test sets and two compounds that were subsequently studied, experimentally demonstrated a moderate predictive power of the M1–M6 QSPR models. Despite the high performance and satisfactory external validation results found for all of the models, we recommend using the M3 and M6 QSPR models for the virtual screening and search for new antioxidants. The M3 and M6 models are based on the combination of the different types of descriptors, which ensures the most objective prognostic estimates of logk_7_.

The satisfactory agreement between the theoretically calculated logk_7_^pred^ values and the experimentally determined logk_7_^exp^ values for the compounds of the test sets TS1, TS2 and antioxidants AO1 and AO2, provides the conclusion that the calculation and selection algorithms for the descriptors, the algorithms of the generation of the regression equations, and their consensus combination implemented in the GUSAR 2019 program allow the correct modeling of the kinetic parameter logk_7_, which is determined experimentally in the model liquid-phase oxidation reactions of organic hydrocarbons.

## 5. Patents

This work was supported by grant No. 19-73-20073 of the Russian Science Foundation.

## Figures and Tables

**Figure 1 molecules-27-06511-f001:**
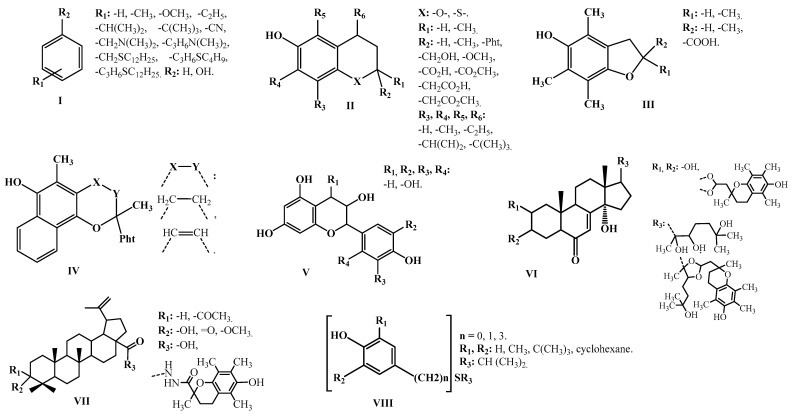
General structural of the formulas of the modeled antioxidant inhibitors (Pht=CH_2_[CH_2_CH_2_CH(CH_3_)CH_2_]_3_H). **I, VIII** (a phenol derivative), **II–V** (chromone derivatives), **VI** (20-hydroxyecdysone derivatives with chroman-2-yl moiety), **VII** (triterpenoids derivatives with chroman-2-yl moiety).

**Figure 2 molecules-27-06511-f002:**
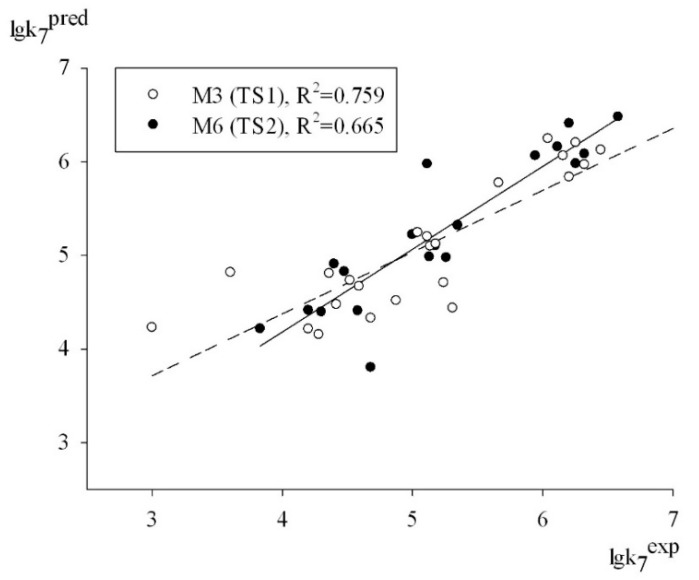
Plot of the predicted vs. the experimental activities based on the M3 and M6 models.

**Figure 3 molecules-27-06511-f003:**
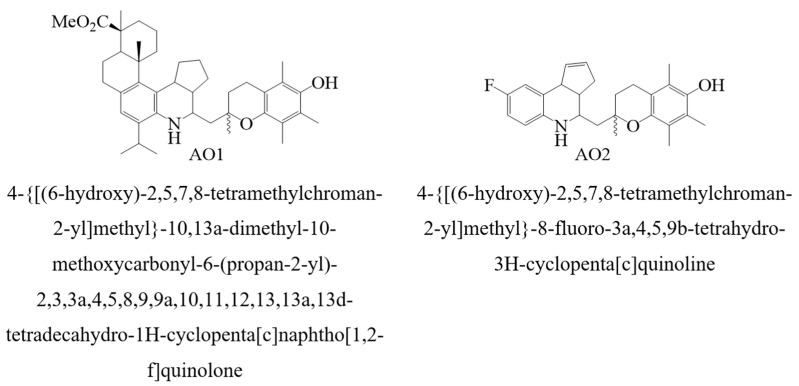
Structures of the compounds designated by AO1 and AO2.

**Figure 4 molecules-27-06511-f004:**
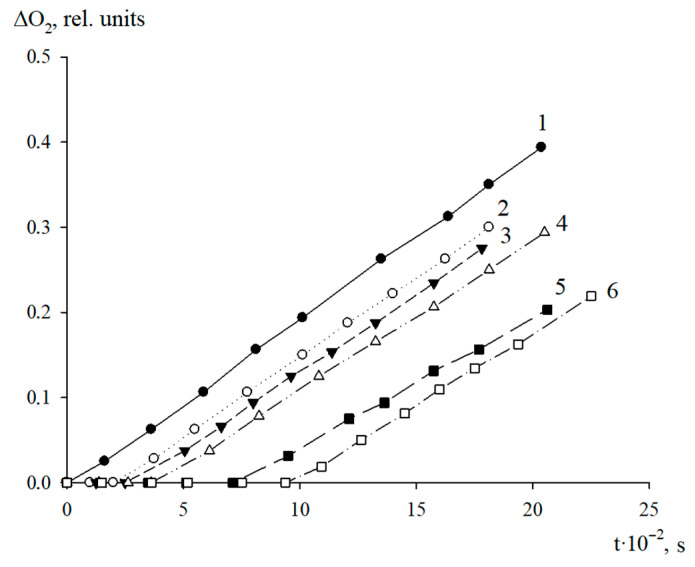
Typical kinetic curves of the oxygen uptake during the oxidation of 1,4-dioxane in the absence (1) and in presence of AO1 taken in concentrations, mol/L: 0.44 × 10^−6^ (2); 1.24 × 10^−6^ (3); 1.88 × 10^−6^ (4); 2.50 × 10^−6^ (5); 3.13 × 10^−6^ (6). T = 348 K, V_i_ = 1 × 10^−7^ M/s.

**Figure 5 molecules-27-06511-f005:**
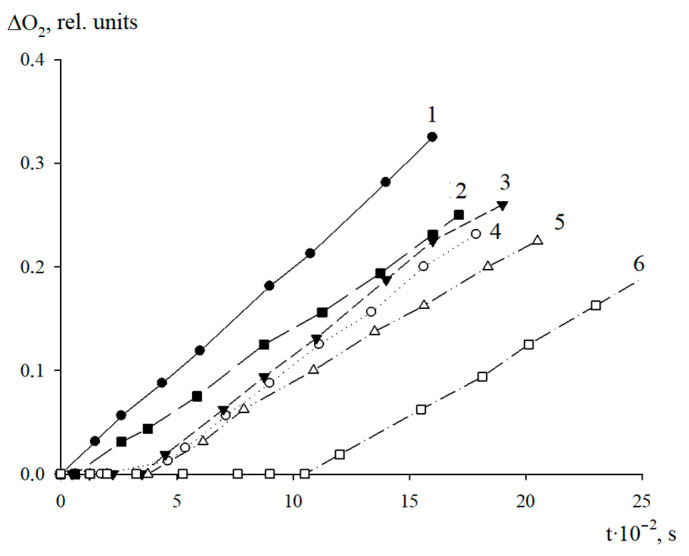
Typical kinetic curves of the oxygen uptake during the oxidation of 1,4-dioxane in the absence (1) and in the presence of AO2 taken in concentrations, mol/L: 0.44 × 10^−6^ (2); 0.94 × 10^−6^ (3); 1.25 × 10^−6^ (4); 1.88 × 10^−6^ (5); 3.13 × 10^−6^ (6). T = 348 K, V_i_ = 1 × 10^−7^ M/s.

**Figure 6 molecules-27-06511-f006:**
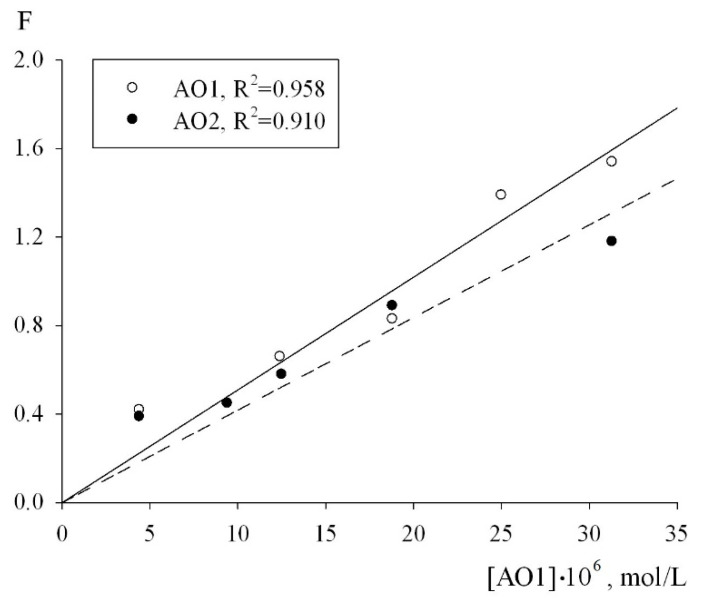
Dependence of the inhibition efficiency parameter on the concentration of AO1 and AO2, V_i_ = 1 × 10^−7^ M/s, T = 348 K.

**Figure 7 molecules-27-06511-f007:**
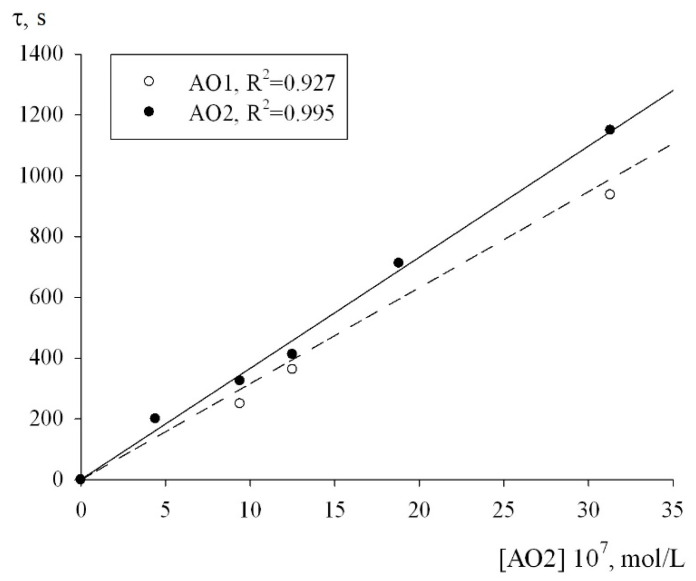
Dependence of the induction period on the injected initial concentration of the inhibitor. T = 348 K, V_i_ = 1 × 10^−7^ M/s.

**Figure 8 molecules-27-06511-f008:**
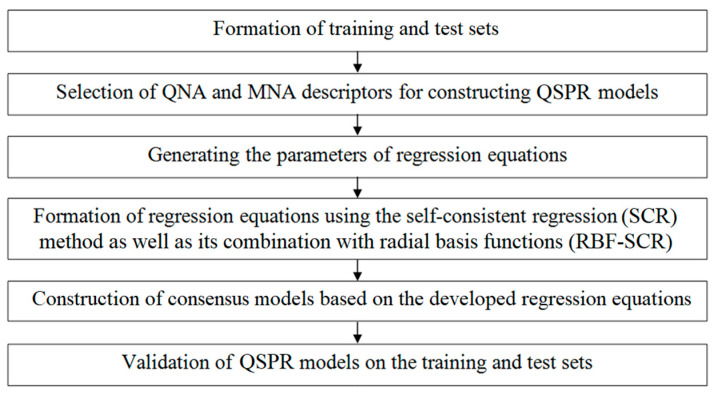
Schematic representation of the GUSAR algorithm.

**Figure 9 molecules-27-06511-f009:**
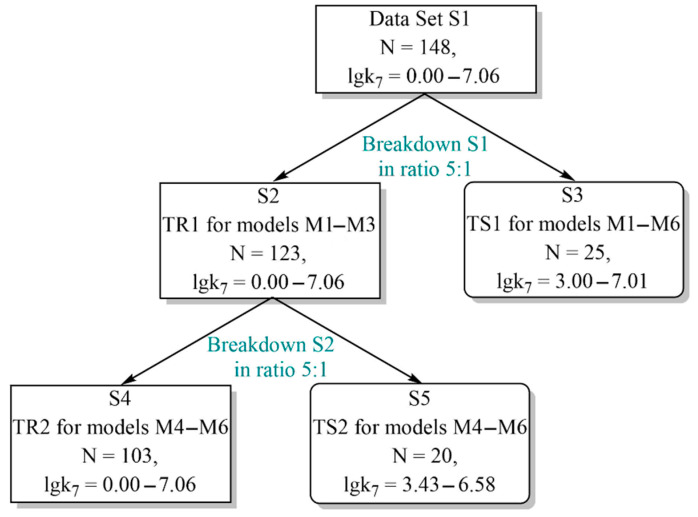
Construction of the training and test sets for the M1–M6 models in the design of the QSPR consensus models (S is set, TR and TS are training and test sets, M is the model, N is the number of compounds included in the corresponding sets and arrays). Designations: (1) S1 is the overall data set; (2) S2 is the training set TR1 for the M1–M3 models; (3) S3 is the external test set TS1 for the M1–M6 models; (4) S4 is the training set TR2 for the M4–M6 models; (5) S5 is the internal test set TS2 for the M4–M6 models.

**Figure 10 molecules-27-06511-f010:**
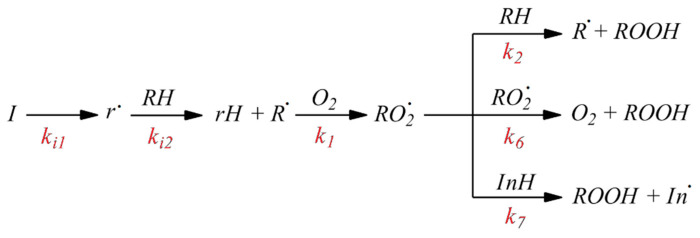
Mechanism of the inhibited radical chain oxidation of organic compounds (I, RH and InH are the initiator, oxidized substrate, and inhibitor, respectively), where I is the initiator of the oxidation process, r^•^ is the radical that was formed upon the decay of the initiator I, RH is the oxidation substrate, R^•^ is the radical that was formed upon the elimination of a hydrogen atom from the substrate molecule by the initiator radical r^•^, RO_2_^•^ is the peroxyl radical formed upon the reaction of the substrate radical R^•^ with an oxygen molecule, InH is antioxidant, In^•^ is the radical formed as a result of the hydrogen atom elimination from the antioxidant molecule by the substrate peroxyl radical RO_2_^•^.

**Table 1 molecules-27-06511-t001:** Statistical parameters and the accuracy of the predicted logk_7_ values of the compounds included in the training sets TR1, TR2 within the M1–M6 consensus models (using Both). ∆logk_7(TR1)_ = ∆logk_7(TR2)_ = 7.057 ^1^.

Training Set	Model	N	N_PM_	$\bar{R_{}^{2}}$	$\bar{F_{}}$	$\bar{SD}$	$\bar{Q_{}^{2}}$	V
**QSPR models based on the QNA descriptors**								
TR1	M1	123	20	0.968	7.675	0.548	0.760	29
TR2	M4	103	20	0.962	7.337	0.587	0.740	24
**QSPR models based on the MNA descriptors**								
TR1	M2	123	20	0.968	7.008	0.550	0.763	29
TR2	M5	103	20	0.964	7.891	0.578	0.756	22
**QSPR models based on both QNA and MNA descriptors**								
TR1	M3	123	320	0.976	8.708	0.512	0.802	28
TR2	M6	103	320	0.973	8.057	0.551	0.787	23

^1^ N is the number of structures in the training set; N_PM_ is the number of regression equations used for the consensus model; $\bar{R_{}^{2}}$ is the determination coefficient calculated for the compounds of TRi; $\bar{Q_{}^{2}}$ is the correlation coefficient calculated for the training set by the cross-validation with the exception of one; $\bar{F_{}}$ is Fisher’s criterion; $\bar{SD}$—standard deviation; V is the number of variables in the final regression equation.

**Table 2 molecules-27-06511-t002:** Validation parameters of the QSPR models estimated using the Xternal Validation Plus 1.2 program based on the experimental and predicted logk_7_ values of the compounds of the internal training sets TR1 and TR2. Δlogk_7(TR1)_ = ∆logk_7(TR2)_ = 7.057 ^1^.

Comments	Prediction Parameters	QSPR Model Used for Predicting logk_7_					
		TR1			TR2		
		M1	M2	M3	M4	M5	M6
Classical metrics (after removing 5% of the data with high residuals)	R^2^	0.9868	0.9849	0.9896	0.9887	0.9850	0.9925
	R^2^_0_	0.9845	0.9837	0.9876	0.9870	0.9839	0.9903
	R^2’^_0_	0.9236	0.9338	0.9317	0.9353	0.9366	0.9364
	$\bar{R_{m}^{2}}$	0.9384	0.9496	0.9454	0.9419	0.9532	0.9414
	∆R^2^_m_	0.0141	0.0149	0.0113	0.0132	0.0146	0.0099
	CCC	0.9916	0.9912	0.9932	0.9932	0.9912	0.9947
Mean absolute error and standard deviation for the test set (after removing 5% of the data with high residuals)	RMSE	0.1090	0.1107	0.0975	0.1128	0.1098	0.0976
	MAE	0.0855	0.0894	0.0765	0.0924	0.0879	0.0773
	SD	0.0679	0.0656	0.0607	0.0650	0.0661	0.0599
	MAE+3SD	0.2892	0.2862	0.2586	0.2873	0.2861	0.2570
Prediction quality	-	Good					
Presence of systematic errors	-	Absent					

^1^ R^2^, R^2^_0_, and R’^2^ are the determination coefficients calculated with and without taking into account the origin; average R^2^_m_ is the averaged determination coefficient of the regression function calculated using the values of determination coefficients on the ordinate axis (R^2^_m_) and on the abscissa axis (R’^2^_m_), respectively; ΔR^2^_m_ is the difference between R^2^_m_ and R’^2^_m_; CCC is the concordance correlation coefficient; MAE is the mean absolute error; SD is the standard deviation.

**Table 3 molecules-27-06511-t003:** Validation parameters of the QSPR models estimated using the Xternal Validation Plus 1.2 program based on the experimental and predicted logk_7_ values of the compounds of test sets TS1 and TS2. ∆logk_7(TR1)_ = ∆logk_7(TR2)_ = 7.057; Δlogk_7(TS1)_ = 4.009; ∆logk_7(TS2)_ = 3.148 ^1^.

Comments	Prediction Parameters	QSPR Model Used for Predicting logk_7_								
		TS1						TS2		
		M1	M2	M3	M4	M5	M6	M4	M5	M6
Classical metrics (after removing 5% of the data with high residuals)	R^2^	0.8204	0.7364	0.7715	0.7696	0.7289	0.7807	0.7765	0.8125	0.8071
	R^2^_0_	0.8115	0.7342	0.7701	0.7621	0.7263	0.7741	0.7739	0.7936	0.8013
	R^2’^_0_	0.5555	0.4652	0.5346	0.4750	0.4466	0.5005	0.6304	0.7650	0.7064
	Q^2^_F1_	0.9538	0.9390	0.9525	0.4750	0.9367	0.9468	0.9567	0.9608	0.9621
	Q^2^_F2_	0.7966	0.7312	0.7627	0.7473	0.7212	0.7656	0.7679	0.7896	0.7969
	$\bar{R_{m}^{2}}$	0.6798	0.6010	0.6726	0.6191	0.5892	0.6332	0.6934	0.7434	0.7353
	∆R^2^_m_	0.1673	0.2191	0.1803	0.2046	0.2249	0.1964	0.1316	0.0319	0.0653
	CCC	0.8763	0.8371	0.8600	0.8433	0.8293	0.8563	0.8775	0.8998	0.8970
Mean absolute error and standard deviation for the test set (after removing 5% of the data with high residuals)	RMSE	0.4133	0.4750	0.4186	0.4606	0.4838	0.4436	0.3870	0.3685	0.3620
	MAE	0.3146	0.3309	0.2986	0.3296	0.3442	0.3129	0.2945	0.2719	0.2740
	SD	0.2740	0.3485	0.3000	0.3289	0.3476	0.3215	0.2580	0.2555	0.2431
	MAE+3SD	1.1367	1.3763	1.1985	1.3164	1.3871	1.2773	1.0684	1.0383	1.0032
Prediction quality	-				Good					
Presence of systematic errors	-				Absent					

^1^ R^2^, R^2^_0_, and R’^2^ are the determination coefficients calculated with and without taking into account the origin; average R^2^_m_ is the averaged determination coefficient of the regression function calculated using the determination coefficients on the ordinate axis (R^2^_m_) and on the abscissa axis (R’^2^_m_), respectively; ∆R^2^_m_ is the difference between R^2^_m_ and R’^2^_m_; CCC is the concordance correlation coefficient; MAE is the mean absolute error; SD is the standard deviation.

**Table 4 molecules-27-06511-t004:** Prediction of logk_7_ for antioxidants AO1 and AO2, based on the M1–M6 models.

Model	Applicability (AD)	Predicted Value of logk_7_^pred^		Experimental Value of logk_7_^exp 1^		Δlogk_7_ ^2^		2RMSEP (95%) ^3^
		AO1	AO2	AO1	AO2	AO1	AO2	
M1	in AD	5.21	5.10	4.64	4.43	0.57	0.67	0.83
M2	in AD	4.79	5.32			0.15	0.89	0.95
M3	in AD	5.17	5.21			0.53	0.78	0.84
M4	in AD	5.25	5.23			0.61	0.80	0.92
M5	in AD	5.07	5.20			0.43	0.77	0.97
M6	in AD	5.19	5.15			0.55	0.72	0.89

^1^ The experimental determination of logk_7_ for compounds AO1 and AO2 is decribed in Section 3; ^2^ ∆logk_7_ = logk_7_^pred^ − logk_7_^exp^; ^3^ The maximum values of the RMSEP were taken; multiplying this criterion by two gives the confidence interval with 95% probability (relative to the predicted value of logk_7_, if the model is correct and the errors are normally distributed, which was observed in our computational experiments) [32].

**Table 5 molecules-27-06511-t005:** Dependence of the initial oxidation rate of ethylbenzene on the concentration of AO1 and AO2; V_i_ = 1·10^−7^ M/s, T = 348 K.

[AO1]·10^6^, mol/L	V_0_·10^6^, M/s	[AO2]·10^6^, mol/L	V_0_·10^6^, M/s
0.00	2.30	0.00	2.36
0.44	1.86	0.44	1.95
1.24	1.66	0.94	1.89
1.88	1.53	1.25	1.77
2.50	1.20	1.88	1.53
3.13	1.13	3.13	1.44

**Table 6 molecules-27-06511-t006:** Statistical characteristics of the training sets TR1, TR2.

Designation of TRi	Code of the Training Set	
	TR1	TR2
N	123	103
logk_7_	3.529	
∆logk_7_	7.057	
**Thresholds used to evaluate the model’s forecast**		
0.10 × ∆logk_7_	0.706	
0.15 × ∆logk_7_	1.059	
0.20 × ∆logk_7_	1.411	
0.25 × ∆logk_7_	1.764	

**Table 7 molecules-27-06511-t007:** Statistical characteristics of test sets TS1, TS2.

Designation of TSi	Code of the Test Set	
	TS1	TS2
N	25	20
$\bar{{\mathrm{lgk}}_{7}}$	5.106	5.117
∆logk_7_	4.009	3.148
**Distribution of the observed response values of test sets TSi around the test mean (in %)**		
$\bar{{\mathrm{lgk}}_{7}}$ ± 0.5, %	32.000	35.000
$\bar{{\mathrm{lgk}}_{7}}$ ± 1.0, %	64.000	70.000
$\bar{{\mathrm{lgk}}_{7}}$ ± 1.5, %	88.000	95.000
$\bar{{\mathrm{lgk}}_{7}}$ ± 2.0, %	96.000	100.000
**Distribution of the observed response values of test sets TSi around the training mean (in %)**		
$\bar{{\mathrm{lgk}}_{7}}$ ± 0.5, %	8.000	10.000
$\bar{{\mathrm{lgk}}_{7}}$ ± 1.0, %	32.000	30.000
$\bar{{\mathrm{lgk}}_{7}}$ ± 1.5, %	44.000	45.000
$\bar{{\mathrm{lgk}}_{7}}$ ± 2.0, %	68.000	70.000

## Data Availability

Not applicable.

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
