# Peer review of "QSPR Modeling and Experimental Determination of the Antioxidant Activity of Some Polycyclic Compounds in the Radical-Chain Oxidation Reaction of Organic Substrates"

_molecules, 2022, doi:10.3390/molecules27196511_

Round 1

Reviewer 1 Report

In this paper six structure-activity models (M1 –M6) are designed to test the antioxidant efficiency of some known molecules with the final aim of proposing a way to find new antioxidants for organic hydrocarbons. Most of the models present high descriptive ability, but low prognostic one. M3 and M6 are the best among all presented models. Besides testing the models against different families of compounds whose antioxidant kinetics are known, the rates of two potential antioxidants are experimentally measured and compared with those calculated with the proposed models.

The work should be published in a journal specialized in statistics, because the technical part of the work is hard to follow for non-specialists and the conclusions are difficult to be used for chemists.

In several places acronyms and symbols are used before their meaning is given. As a general rule, the first time an acronym appears, its meaning has to be declared. Afterwards, the acronym can be used without its description.

Line 114. Authors refer to previous work but they do not provide the corresponding references. They are given much later with numbers 59-70.

Figure 1 could be improved. It displays the scaffolds of the molecules used to test the statistical models but much more information about the 148 molecules can be given just by indicating the different species/atoms represented by R1, X, Y. Besides, compounds may be arranged in chemical families whose name should also be provided.

Line 323. Add after Mi (i = 1 - 6).

Author Response

Reviewer 1

 The authors would like to express their sincere gratitude to the reviewer for his detailed analysis of the manuscript material. The edits we have made to the text, as requested by the reviewer, are highlighted in green.

Response to Reviewer 1 Comments

1) The work should be published in a journal specialized in statistics, because the technical part of the work is hard to follow for non-specialists and the conclusions are difficult to be used for chemists.

We do not fully agree with this comment.

In fact, this work addresses an urgent problem in physical organic chemistry: the quantitative study of the structure-property relationship. As the last in the present work are quantitative characteristics of the antioxidant efficiency of a number of polycyclic organic compounds. This physico-chemical task is discussed, along with the statistical characteristics, in the introduction, the text of the paper and in the conclusions. At the same time, we considered it necessary, in the text of the article and in the conclusions, to pay attention to the description of descriptor calculation, ideology of their selection for model building, description of statistical characteristics, based on which descriptive and predictive abilities of QSPR models are evaluated. Please note that articles of similar content have been published in Molecules, both by us and by other authors. For example,

1) Butkiewicz, M.; Lowe, E.W.Jr.; Mueller, R.; Mendenhall, J.L.; Teixeira, P.L.; Weaver, C.D.; Meiler, J. Benchmarking Ligand-Based Virtual High-Throughput Screening with the PubChem Database. Molecules 2013, 18, 735-756. doi: 10.3390/molecules18010735

2) Lozano, N.B.H.; Oliveira, R.F.; Weber, K.C.; Honorio, K.M.; Guido, R.V.; Andricopulo, A.D.; Silva, A.B.F.D. Identification of Electronic and Structural Descriptors of Adenosine Analogues Related to Inhibition of Leishmanial Glyceraldehyde-3-Phosphate Dehydrogenase. Molecules 2013, 18, 5032-5050. doi: 10.3390/molecules18055032

3) De Souza, S.D.; De Souza, A.M.T.; De Sousa, A.C.C.; Sodero, A.C.R.; Cabral, L.M.; Albuquerque, M.G.; Castro, H.C.; Rodrigues, C.R. Hologram QSAR Models of 4-[(Diethylamino)methyl]-phenol Inhibitors of Acetyl/Butyrylcholinesterase Enzymes as Potential Anti-Alzheimer Agents. Molecules 2012, 17, 9529-9539. doi: 10.3390/molecules17089529

4) Khairullina, V.; Safarova, I.; Sharipova, G.; Martynova, Y.; Gerchikov, A. QSAR Assessing the Efficiency of Antioxidants in the Termination of Radical-Chain Oxidation Processes of Organic Compounds. Molecules 2021, 26, 421, doi:10.3390/molecules2602042

Unfortunately, we cannot completely remove from the manuscript the description of the principles of GUSAR2019 and the analysis of statistical characteristics of M1-M6 models due to comments and requests of two other reviewers, as well as based on reviewers' comments in the publication of our earlier works in chemically oriented journals, including this journal.

The fact is that GUSAR2019 is a new intellectual product. It is currently undergoing extensive testing on a variety of biological activities. It differs fundamentally from foreign and domestic equivalents in terms of the ideology of descriptor calculation, their selection, and the construction of regression equations and QSAR/QSPR models based on them.

It is these factors that have prompted us to provide a detailed description of the GUSAR2019 programme, its underlying algorithms, and a detailed analysis of most of the generally accepted statistical characteristics as the evidence base for its applicability.

2) In several places acronyms and symbols are used before their meaning is given. As a general rule, the first time an acronym appears, its meaning has to be declared. Afterwards, the acronym can be used without its description.

The authors agree with the comment. Corrections have been made to the text of the article accordingly.

3) Line 114. Authors refer to previous work but they do not provide the corresponding references. They are given much later with numbers 59-70.

The authors agree with the comment. Corresponding edits have been made to the text of the article and reference numbers [62,63-66] (p. 3).

4) Figure 1 could be improved. It displays the scaffolds of the molecules used to test the statistical models but much more information about the 148 molecules can be given just by indicating the different species/atoms represented by R1, X, Y.

The authors agree with the comment. Corrections have been made accordingly (see Figure 1).

5) Besides, compounds may be arranged in chemical families whose name should also be provided.

The authors of the article fully agree with the comment. Corresponding edits have been made to the text of the article. The corrections we have made to the text, as requested by the reviewer, are highlighted in green.

6) Line 323. Add after Mi (i = 1 - 6).

 Corresponding edits have been made to the text of the article. The corrections we have made to the text, as requested by the reviewer, are highlighted in green.

Reviewer 2 Report

In the manuscript entitled “Determination of Antioxidant Activity of Certain Mono- and Polycycles in the Radical Chain Oxidation Reaction of 1,4-Dioxane by QSPR-Modeling”, the authors used the descriptor calculation algorithms implemented in the GUSAR2019 software to model the kinetic parameters of the oxidation reactions of organic hydrocarbons. They developed six statistically significant QSPR models, which have satisfactory accuracy in predicting Reaction rate constants. Their theoretically predictions of the antioxidant activity of the investigated compounds are based and correlated to the experimentally determinated antioxidativity of the 1,4-dioxane.

The writing style is very clear and concise. The results are presented correctly.

With everything mentioned in mind, I suggest publishing this manuscript after minor changes:

1.       The title should better describe the main topic and essence of the research. Accordingly, I suggest changing the title.

2.       Explain what the marks R1, R2, ....., R5, R6, than Pht, X and Y, represent in Figure 1.

3.       In line 91, it is better to replace the term “schools” with the term “scientific groups.

4.       In the Supplementary Material, the references are marked in green. This mark should be removed.

5.       The English language should be checked and corrected.

6.       I suggest the citation of following references:

-          S. Jeremić, S. Radenković, M. Filipović, M. Antić, A. Amić, Z. Marković, Importance of hydrogen bonding and aromaticity indices in QSAR modeling of the antioxidative capacity of selected (poly)phenolic antioxidants, Journal of Molecular Graphics and Modelling (2017) 72 (00) 240–245.

-           Z. Marković, M. Filipović, N. Manojlović, A. Amić, S. Jeremić, D. Milenković. QSAR of the free radical scavenging potency of selected hydroxyanthraquinones, Chemical Papers (2018) 72 (11) 2785–2793.

Author Response

Reviewer 2

 The authors would like to express their sincere gratitude to the reviewer for his detailed analysis of the manuscript material. The edits we have made to the text, as requested by the reviewer, are highlighted in green.

Response to Reviewer 2 Comments

1) The title should better describe the main topic and essence of the research. Accordingly, I suggest changing the title.

The authors fully agree with the reviewer.

In the revised version of the manuscript, the title has been changed. Now, in our opinion, it more accurately reflects the essence of the research performed.

2) Explain what the marks R1, R2, ....., R5, R6, than Pht, X and Y, represent in Figure 1.

The authors agree with the comment. Relevant comments have been added to the text of the manuscript (see Figure 1).

3) In line 91, it is better to replace the term “schools” with the term “scientific groups.”

The authors agree with the comment. Relevant comments have been added to the text of the manuscript.

4) In the Supplementary Material, the references are marked in green. This mark should be removed.

The authors agree with the comment. Corrections have been made to the Supplementary Material accordingly.

5) The English language should be checked and corrected.

The authors agree with the comment. Corresponding edits to the text have been made.

6) I suggest the citation of following references:

-          S. Jeremić, S. Radenković, M. Filipović, M. Antić, A. Amić, Z. Marković, Importance of hydrogen bonding and aromaticity indices in QSAR modeling of the antioxidative capacity of selected (poly)phenolic antioxidants, Journal of Molecular Graphics and Modelling (2017) 72 (00) 240–245.

-           Z. Marković, M. Filipović, N. Manojlović, A. Amić, S. Jeremić, D. Milenković. QSAR of the free radical scavenging potency of selected hydroxyanthraquinones, Chemical Papers (2018) 72 (11) 2785–2793.

Corresponding edits to the text have been made [21,22].

Reviewer 3 Report

This work is devoted to the study of the antioxidant activity of some mono- and polycycles in the radical chain reaction of 1,4-dioxane oxidation using QSPR modeling. This article is written in a clear and accessible language. The abundance of data and the quality of the presentation help to understand the goals achieved by the authors. However, it is desirable to improve the following points:

1. "...formulas I–XI (Fig. 1)". In Abstract, it is customary to do transcripts and not to make references to figures in the text. You can make some generalizations. Those who need it will refer to the text of the article itself.

2. In the introduction, state more clearly the practical significance of this work.

3. What is the advantage of the QSPR method for calculating such systems?

4. The conclusions are too voluminous. Make them more concise.

5. Please cite article: 10.3390/ijms23031602.

Author Response

Reviewer 3

 The authors would like to express their sincere gratitude to the reviewer for his detailed analysis of the manuscript material. The edits we have made to the text, as requested by the reviewer, are highlighted in green.

Response to Reviewer 3 Comments

1) "...formulas I–XI (Fig. 1)". In Abstract, it is customary to do transcripts and not to make references to figures in the text. You can make some generalizations. Those who need it will refer to the text of the article itself.

The authors agree with the comment. Relevant comments have been added to the text of the manuscript.

2) In the introduction, state more clearly the practical significance of this work.

The authors agree with the comment. Relevant comments have been added to the text of the manuscript.

3) What is the advantage of the QSPR method for calculating such systems?

The answer to this question has been added to the Introduction section.

4) The conclusions are too voluminous. Make them more concise.

The authors agree with the comment. In the corrected version of the manuscript, we have tried to shorten the conclusions of the completed study. Corresponding edits are highlighted in green.

5) Please cite article: 10.3390/ijms23031602.

Corresponding edits to the text have been made [23].

Round 2

Reviewer 1 Report

Previous comments have been considered in the manuscript or correctly justified by authors.

There are just some last mistakes in Figure 1: In structure I no R2 substituents are present. The same happens with R5 in structure V. However, in Structure II there is no information for R6.

Author Response

Reviewer 1

 The authors would like to express their sincere gratitude to the reviewer for his detailed analysis of the manuscript material. The edits we have made to the text, as requested by the reviewer, are highlighted in green.

Response to Reviewer 1 Comments

  1. There are just some last mistakes in Figure 1: In structure I no R2 substituents are present. The same happens with R5 in structure V. However, in Structure II there is no information for R6.

The authors agree with the comment. Appropriate corrections have been made to Figure 1.
